# MYC regulates ductal-neuroendocrine lineage plasticity in pancreatic ductal adenocarcinoma associated with poor outcome and chemoresistance

Amy S. Farrell[1], Meghan Morrison Joly[1], Brittany L. Allen-Petersen[1], Patrick J. Worth[2], Christian Lanciault[3], David Sauer[3], Jason Link[1,4], Carl Pelz[4,5], Laura M. Heiser[6], Jennifer P. Morton[7], Nathiya Muthalagu[7], Megan T. Hoffman[8], Sara L. Manning[9], Erica D. Pratt[9], Nicholas D. Kendsersky[1], Nkolika Egbukichi[1], Taylor S. Amery[4], Mary C. Thoma[1], Zina P. Jenny[1], Andrew D. Rhim[9], Daniel J. Murphy [7,10], Owen J. Sansom [7], Howard C. Crawford[8], Brett C. Sheppard[2,4,11] & Rosalie C. Sears[1,4,11]

Intratumoral phenotypic heterogeneity has been described in many tumor types, where it can contribute to drug resistance and disease recurrence. We analyzed ductal and neuroendocrine markers in pancreatic ductal adenocarcinoma, revealing heterogeneous expression of the neuroendocrine marker Synaptophysin within ductal lesions. Higher percentages of Cytokeratin-Synaptophysin dual positive tumor cells correlate with shortened disease-free survival. We observe similar lineage marker heterogeneity in mouse models of pancreatic ductal adenocarcinoma, where lineage tracing indicates that Cytokeratin-Synaptophysin dual positive cells arise from the exocrine compartment. Mechanistically, MYC binding is enriched at neuroendocrine genes in mouse tumor cells and loss of MYC reduces ductal-neuroendocrine lineage heterogeneity, while deregulated MYC expression in KRAS mutant mice increases this phenotype. Neuroendocrine marker expression is associated with chemoresistance and reducing MYC levels decreases gemcitabine-induced neuroendocrine marker expression and increases chemosensitivity. Altogether, we demonstrate that MYC facilitates ductal-neuroendocrine lineage plasticity in pancreatic ductal adenocarcinoma, contributing to poor survival and chemoresistance.

---

[1] Department of Molecular and Medical Genetics, Oregon Health and Science University, 3181 S.W. Sam Jackson Park Road, Portland, OR 97239, USA. [2] Department of Surgery, Oregon Health and Science University, 3181 S.W. Sam Jackson Park Road, Portland, OR 97239, USA. [3] Department of Pathology, Oregon Health and Science University, 3181 S.W. Sam Jackson Park Road, Portland, OR 97239, USA. [4] Brenden-Colson Center for Pancreatic Care, Oregon Health and Science University, 3181 S.W Sam Jackson Park Road, Portland, OR 97239, USA. [5] Computational Biology, Oregon Health and Science University, 3181 S.W. Sam Jackson Park Road, Portland, OR 97239, USA. [6] Department of Biomedical Engineering and OHSU Center for Spatial Systems Biomedicine, Oregon Health and Science University, 3181 S.W. Sam Jackson Park Road, Portland, OR 97239, USA. [7] Cancer Research UK, Beatson Institute, Switchback Road, Bearsden, Glasgow G61 1BD, UK. [8] Department of Molecular and Integrative Physiology, University of Michigan, 7744 MS II, 1137 E. Catherine St., Ann Arbor, MI 48109, USA. [9] Department of Gastroenterology, Hepatology and Nutrition and Zayed Center for Pancreatic Cancer Research, University of Texas M.D. Anderson Cancer Center, Unit 1466, 1515 Holcombe Blvd, Houston, TX 77030, USA. [10] Institute of Cancer Sciences, University of Glasgow, University Avenue, Glasgow G12 8QQ, UK. [11] Knight Cancer Institute, Oregon Health and Science University, 3181 S.W. Sam Jackson Park Road, Portland, OR 97239, USA. Amy S. Farrell and Meghan Morrison Joly contributed equally to this work. Correspondence and requests for materials should be addressed to R.C.S. (email: searsr@ohsu.edu)

D espite focused research efforts, advanced pancreatic ductal adenocarcinoma (PDA) remains almost uniformly fatal due, in large part, to late diagnosis when few effective treatment options are available. Unfortunately, current standard of care chemotherapeutic agents only marginally improve survival time for these patients[1], underscoring the need for an increased understanding of the factors responsible for therapeutic resistance.

Cell plasticity is a key driver of intratumoral phenotypic heterogeneity, which is thought to contribute to drug resistance and poor outcome in multiple tumor types[2–4]. Increasing evidence suggests that tumor cells with the capacity for phenotypic plasticity may enter states that allow their survival in response to certain selective pressures within their microenvironment. Indeed, studies indicate that drug resistance and disease recurrence in prostate adenocarcinoma (PCA) and non-small cell lung

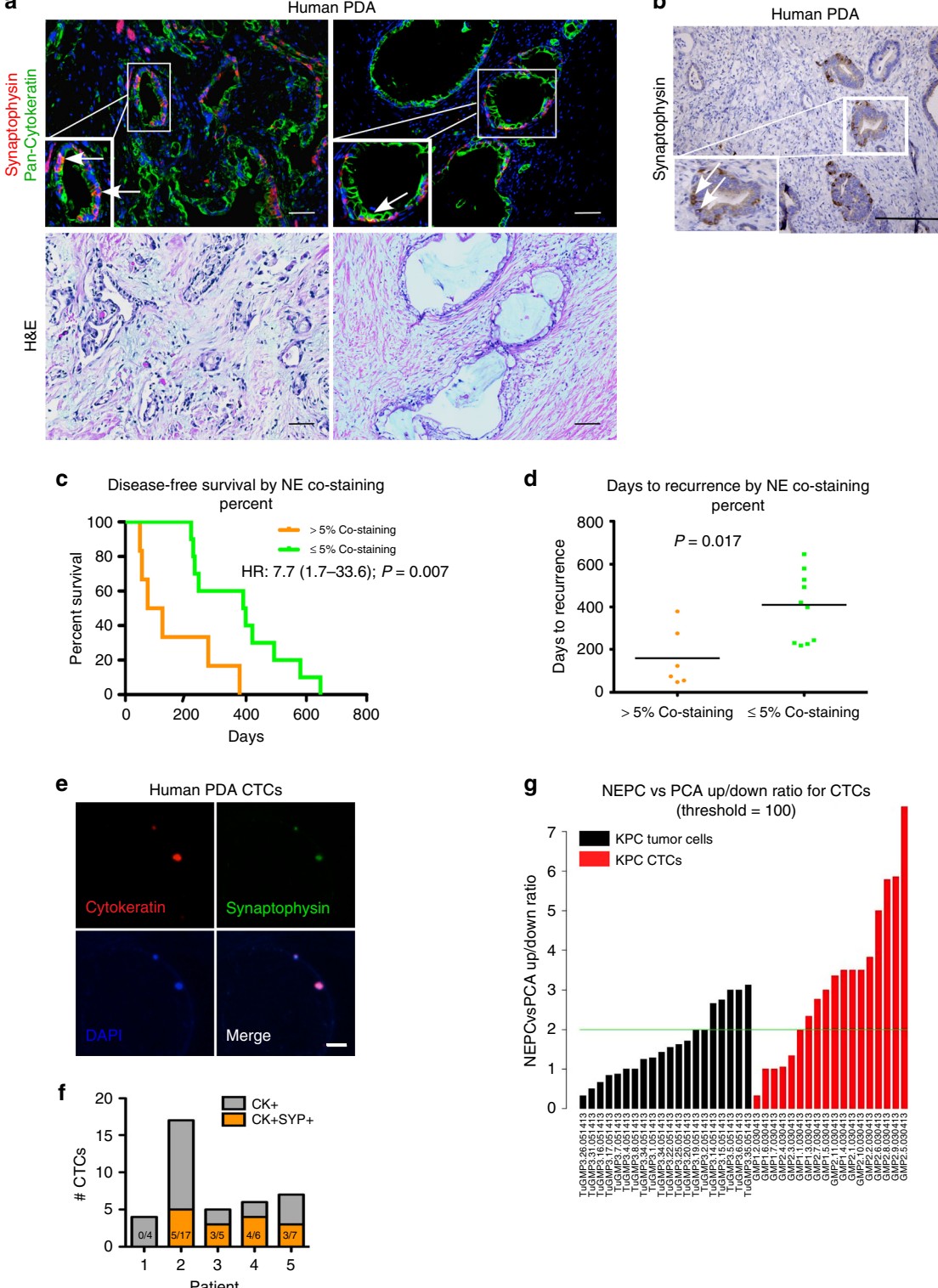

cancer (NSCLC) can involve neuroendocrine transdifferentiation of epithelial tumor cells. For example, in response to androgen-deprivation therapy, prostate tumor cells begin to exhibit a neuroendocrine phenotype and express neuroendocrine markers, such as Synaptophysin (SYP)[5] and Chromogranin A[6] (ChgA), and the degree of prostate tumor cells with neuroendocrine differentiation features correlates with tumor progression and poor prognosis in patients[7, 8]. Additionally, therapeutic pressure can drive conversion of NSCLC to high-grade small cell lung cancer (SCLC) with neuroendocrine morphology[9]. Thus, the existence of tumor cells with neuroendocrine differentiation features may be an important mechanism contributing to aggressive disease and therapeutic resistance. A recent publication identified a subset of pancreatic intraepithelial neoplasia (PanIN)-associated neuroendocrine cells that interacted with sensory neurons to promote PanIN progression and formation[10]. However, further studies are necessary to determine whether ductal-neuroendocrine lineage plasticity plays a role in PDA aggressiveness and therapeutic resistance.

In prostate cancer, the N-MYC oncoprotein plays a critical role in driving therapy refractory neuroendocrine prostate cancer (NEPC) and N-MYC overexpression can induce both PCA and NEPC from a common epithelial precursor[11–13, 27]. Further, c-MYC, in cooperation with Pim1 kinase, can drive invasive prostate carcinoma with neuroendocrine differentiation[14]. Previous studies have defined a RAS-regulated signaling pathway that activates c-MYC (MYC) by phosphorylation at Serine 62, which increases its protein half-life and enhances MYC's transcriptional activity[15–20]. We have demonstrated that this post-translationally modified MYC is highly expressed in pancreatic cancer[21]. In addition, MYC expression increases upon KRAS expression in the mouse pancreas[22]. As PDA is almost universally driven by oncogenic KRAS and given MYC's reported role in neuroendocrine differentiation of PCA, KRAS-induced activation of MYC may contribute to cell lineage plasticity in PDA, that could drive a neuroendocrine differentiation phenotype.

Here, we reveal intratumoral cell lineage heterogeneity in human PDA, where we observe Cytokeratin (CK)-positive epithelial tumor cells expressing neuroendocrine markers. We report that higher percentages of CK-SYP co-expressing cells are significantly associated with decreased disease-free survival in patients. Tumor cells capable of expressing SYP survive passaging through mice in patient-derived xenograft (PDX) models, supporting their tumor origin. Furthermore, we observe similar lineage marker heterogeneity in mouse models of PDA, where in vivo lineage tracing experiments indicate the acinar origin of ductal SYP-expressing cells. Using a novel mouse model of PDA combining low-level deregulated MYC expression with mutant KRAS (KMC), we find that deregulated MYC expression decreases survival and drives ductal-neuroendocrine lineage plasticity, while loss of one copy of *Myc* in the LSL-*Kras*^*G12D*; *P53*^*R172H/+*;*Pdx1-Cre* (KPC) mouse model suppresses ductal-neuroendocrine cells. Further, neuroendocrine marker expression is associated with chemoresistance in human PDA cells, and

reducing MYC expression suppresses gemcitabine-induced neuroendocrine marker expression and increases gemcitabine sensitivity. Taken together, these studies indicate that ductal-neuroendocrine lineage plasticity, regulated in part by MYC, is associated with therapeutic resistance and poor outcome in PDA.

## Results

**Ductal-neuroendocrine lineage plasticity in human PDA**. We analyzed the expression of ductal (CK) and neuroendocrine (SYP) markers in human PDA and observed heterogeneous expression of the neuroendocrine marker SYP, including both discrete SYP positive cells and CK-SYP dual positive cells (Fig. 1a – white arrows). Similar to what has been described in PCA with focal neuroendocrine differentiation[8], we did not observe any obvious morphological changes in the ductal-associated SYP positive cells (Fig. 1a, b). To investigate the clinical relevance of CK-SYP dual positive cells in PDA, we stained tumors for CK and SYP from a cohort of surgically resected patients with R0 status and no evidence of disease following surgery, subsequently treated with standard of care, adjuvant gemcitabine (Supplementary Table 1). Staining was scored as less than 5% or greater than 5% CK-SYP co-staining. We found that higher co-staining (>5%) correlated significantly with shortened disease-free survival (Fig. 1c), with the high co-staining cohort recurring, on average, 297 days earlier than the low co-staining cohort (Fig. 1d). Multivariate analysis demonstrated that clinical stage, grade and age of the patient were not confounding variables in this analysis and thus, high CK-SYP co-staining is an independent risk factor for poor outcomes in PDA. Similar CK-SYP staining patterns were observed in surgically resected tumors from an additional cohort of PDA patients where we found that 3/10 tumors analyzed displayed greater than 5% CK-SYP co-positive cells (Supplementary Table 2), although recurrence data for this cohort is not yet complete. We also analyzed circulating tumor cells (CTCs) from patients with PDA for neuroendocrine marker expression and found co-staining of CK and SYP (Fig. 1e) in 4 of 5 patients assessed with a frequency ranging from 30 to 66% (Fig. 1f). Using a well controlled, single-cell RNA-sequencing (RNA-seq) data set[23] generated from mouse CTCs and their matched GFP-labeled parental tumor cells, we assessed neuroendocrine gene expression utilizing a gene signature distinguishing neuroendocrine prostate cancer (NEPC) from prostate adenocarcinoma (PCA) (hereafter referred to as NEPC gene signature)[11]. We detected a > 2-fold enrichment in NEPC up genes vs. PCA up genes in 66% (12 out of 18) of the CTCs analyzed compared to 20% (5 out of 20) of single-cell counterparts from the parental tumor (Fig. 1g). Taken together, these data suggest that the presence of tumors cells with neuroendocrine differentiation features is correlated with poorer survival and a shortened time to recurrence in PDA, consistent with the presence of neuroendocrine markers and expressed genes in CTCs and the idea that ductal-neuroendocrine lineage plasticity may contribute to aggressive disease.

**Fig. 1** Ductal-neuroendocrine plasticity is associated with decreased survival. **a** Ductal-neuroendocrine lineage plasticity is observed in human PDA. Human PDA samples were stained with SYP and Pan-CK by IF. Co-staining cells are marked with white arrows. H&E from the same tumor sample are shown below. Scale bars indicate 100 μM. **b** Human PDA sample was stained with SYP by IHC. Examples of ductal-associated SYP positive cells are marked with white arrows. Scale bars indicate 100 μM. **c, d** Ductal-neuroendocrine lineage plasticity is associated with poor PDA patient survival. Human PDA samples were stained for SYP and Pan-CK and the total percentage of tumor epithelium co-staining for CK-SYP across entire sections was graded as ≤5% or > 5%, and correlated with disease-free survival **c** and time-to-recurrence **d** as described in the Methods section. For **c**, $P = 0.007$, log-rank test. For **d**, $P = 0.017$, one-tailed Mann Whitney test. **e, f** Ductal-neuroendocrine lineage plasticity can be detected in circulating tumor cells (CTCs) from patients with PDA. CTCs from patients with PDA were stained for PanCK and SYP **e** and the number of SYP positive or CK-SYP dual positive cells was quantified for each patient **f**. Scale bar indicates 25 μM. **g** Analysis of NEPC gene signature in mouse CTCs (red bars) and matched single parental tumor cells (black bars) from KPC mice (GSE51372). Samples were ranked based on ratios of NEPC UP vs. PCA UP (NEPC down) gene expression as described in the Methods section

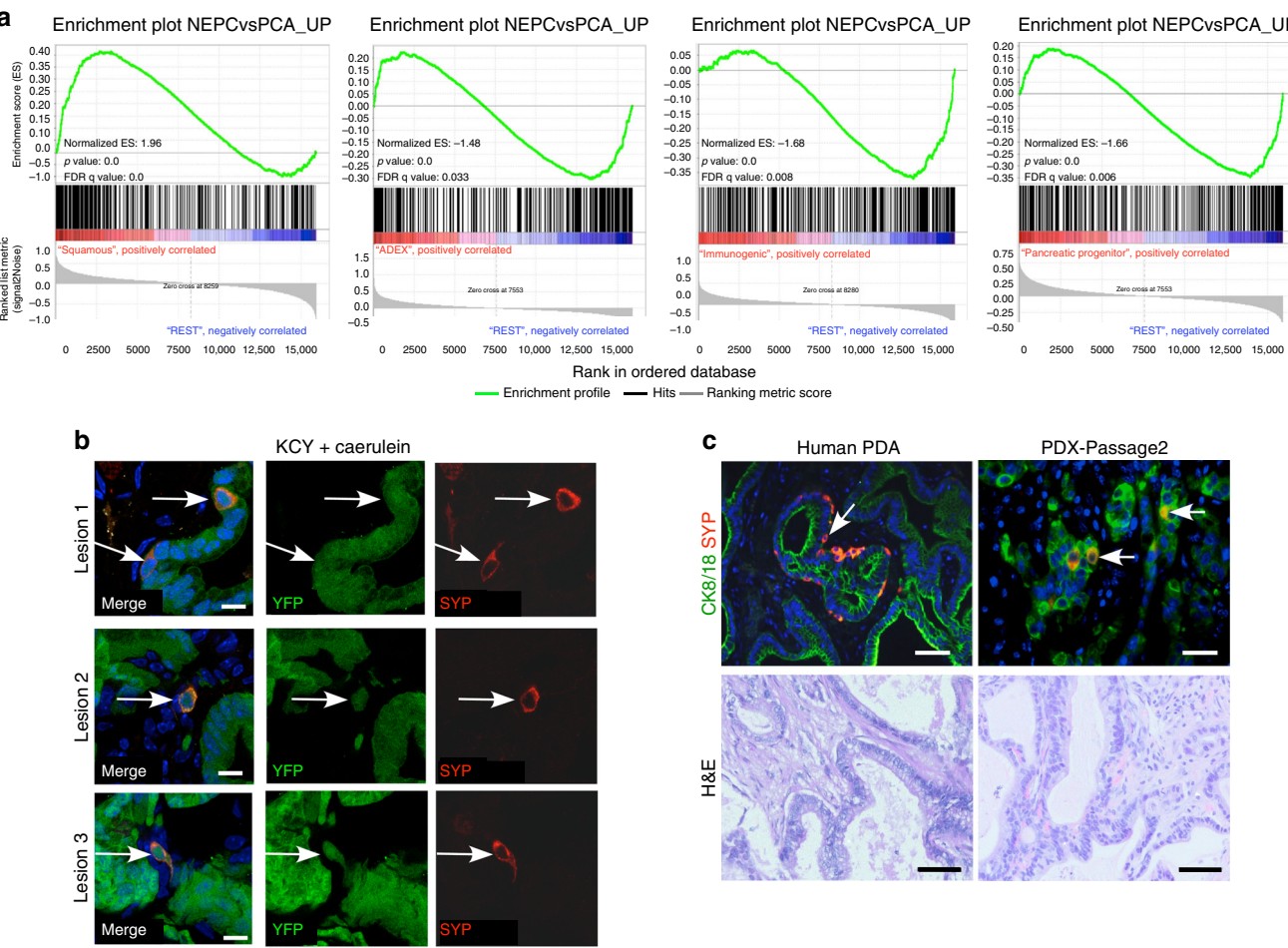

**Fig. 2** Neuroendocrine differentiation can arise from acinar cells. **a** Gene set enrichment analysis (GSEA) demonstrating that genes upregulated in NEPC vs. PCA[11] are enriched in the squamous subtype of pancreatic cancer, compared to all other individual subtypes ("REST"). **b** Pancreatic tissue from KCY mice was stained for SYP and YFP. SYP-YFP dual positive tumor cells are marked with white arrows. Scale bar indicates 10 μM. **c** CK-SYP dual positive cells are observed in serial-passaged patient-derived xenografts (PDXs). Both the primary patient sample and tissue after two passages through mice were stained with CK8/18 and SYP. Examples of co-stained cells are marked with white arrows. Corresponding H&E images are shown below. Scale bars indicate 100 μM

**Neuroendocrine gene expression in poor outcome PDA subtype**. PDA is a heterogeneous disease and recent molecular profiling suggests that several molecular subtypes of PDA may exist. These include the squamous/quasi-mesenchymal, aberrantly differentiated endocrine exocrine (ADEX), and pancreatic progenitor and immunogenic, which together make up the previously identified classical (CL) subtype, with the squamous/quasi-mesenchymal subtype showing significantly worse overall survival[24, 25]. We asked whether the NEPC gene signature was enriched in any of the PDA subtypes using published RNAseq data from 96 PDA specimens[24]. GSEA indicated that NEPC genes were highly enriched in the squamous/quasi-mesenchymal subtype compared to the other subtypes ("REST" - Fig. 2a). We observed negative enrichment of NEPC genes in each of the other subtypes compared to REST (Fig. 2a) and conversely, genes upregulated in PCA vs. NEPC were de-enriched in the squamous/quasi-mesenchymal subtype vs. all other subtypes (Supplementary Fig. 1a), suggesting that NEPC genes are associated with the squamous tumors that show overall poor survival.

**Neuroendocrine differentiation can arise from acinar cells**. To examine the origin of the dual CK + SYP + cells, we utilized in vivo lineage tracing in KCY mice ($Kras^{G12D}$;Rosa-LSL-YFP;

$Ptf1a$-Cre$^{ER}$), which express $Kras^{G12D}$ and YFP in response to Cre-mediated recombination. In these studies, Cre was activated with tamoxifen at 8 weeks, when Ptf1a-driven expression is restricted to acinar and centro-acinar cells, followed by caerulein-induced pancreatitis to accelerate transformation. Following acinar-to-ductal metaplasia, these mice develop PanIN and later PDA such that ductal lesions are YFP positive[26], allowing us to assess the origin of SYP + cells during early progression. We observed SYP-YFP dual positive cells in the ductal lesions of KCY mice (Fig. 2b), demonstrating that these ductal-associated SYP + cells do not arise from the endocrine compartment, but rather are exocrine in origin. Furthermore, we analyzed PDA patient-derived xenografts (PDXs) for SYP and CK expression. We detected CK + SYP + cells in serial-passaged PDX tumor tissue (up to at least 6 passages, passage 2 shown-Fig. 2c) that co-stained with the human Ku80 protein (Supplementary Fig. 1b), indicating that these dual positive cells in human PDA are tumor cell-derived, since normal islets would not be expected to survive the time it takes for PDXs to establish nor propagate in serial passage.

**MYC facilitates ductal-neuroendocrine plasticity in PDA**. Several studies have demonstrated a role for MYC in cellular plasticity, and both c-MYC and N-MYC play a functional role in

neuroendocrine transdifferentiation of prostate cancer cells[11–14], [27]. Therefore, we investigated whether c-MYC contributes to similar ductal-neuroendocrine lineage plasticity in PDA. We stained pancreatic tumors from end stage LSL-Kras$^{G12D}$;P53$^{R172H/+}$;Pdx1-Cre (KPC) and KPC Myc$^{fl/+}$ mice for SYP and CK (Fig. 3a). KPC Myc$^{fl/+}$ mice have reduced PDA progression

(Supplementary Fig. 2a) and increased survival[28]. Compared to KPC mice, loss of one copy of c-Myc significantly reduced the percentage of CK-SYP dual positive cells in ductal lesions (Fig. 3b). We next interrogated MYC ChIP-seq data from KPC PDA tumor cells[28] and found enrichment in MYC DNA binding peaks within 1000 bp of genes which were upregulated in the

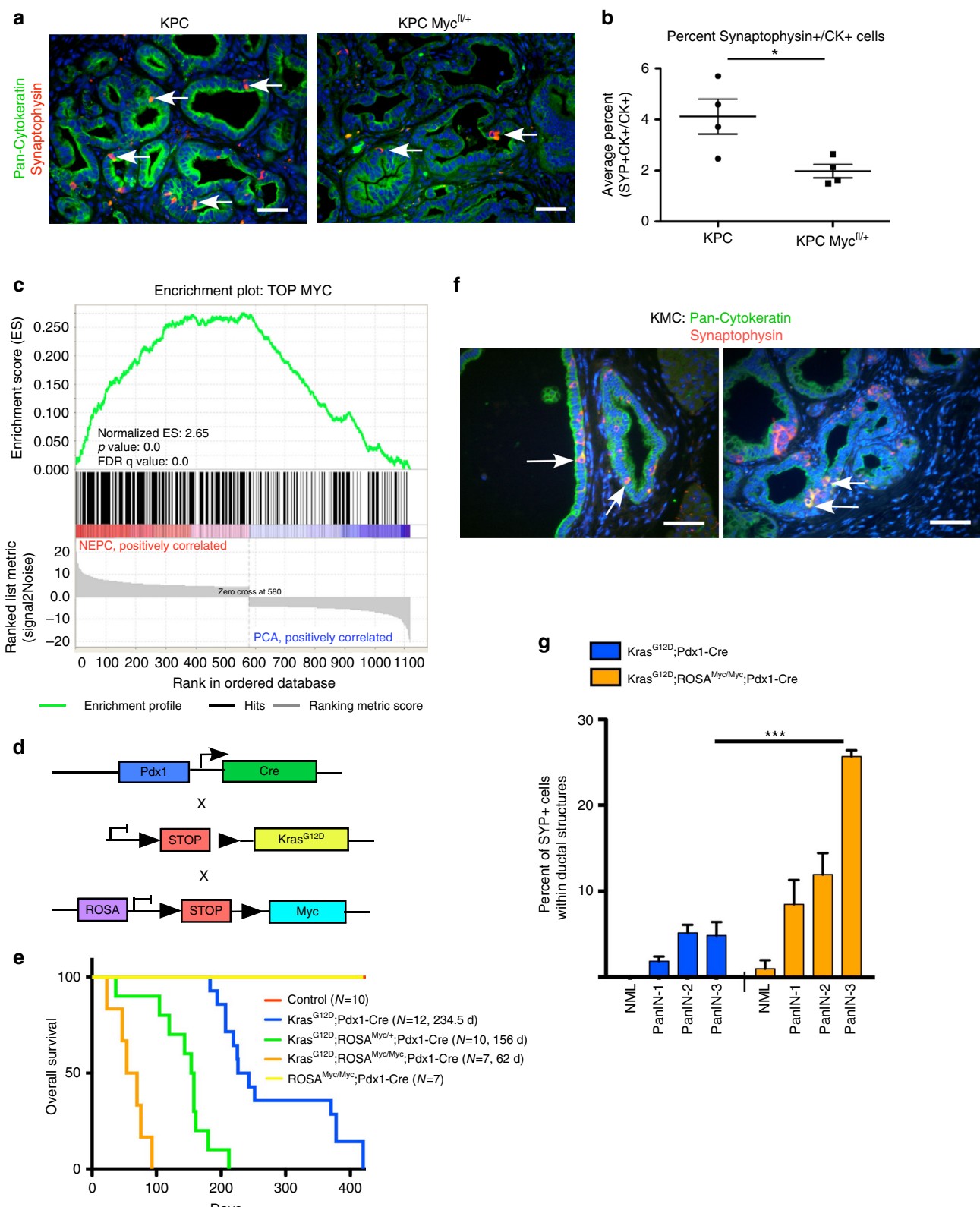

NEPC gene signature compared to genes upregulated in PCA (Fig. 3c), supporting a role for MYC in regulating the ductal-neuroendocrine lineage plasticity in PDA.

MYC DNA binding is enhanced by phosphorylation at a conserved residue, Serine 62 (S62) and work from our lab, and other labs, has revealed that RAS signaling induces the phosphorylation of MYC at S62, which is increased in PDA[15, 16, 19–21, 29]. Therefore, we developed a novel mouse model of PDA driven by KRAS and low, deregulated MYC expression and investigated the degree of intratumoral ductal-neuroendocrine lineage heterogeneity. We crossed our ROSA-LSL-*Myc* mice, which conditionally express physiological levels of MYC[30], with LSL-*Kras*[G12D];Pdx1-*Cre* (KC)[31] mice, to generate LSL-*Kras*[G12D];ROSA-LSL-*Myc*;Pdx1-*Cre* (*Kras*[G12D]ROSA[Myc/Myc];Pdx-Cre or KMC) mice (Fig. 3d). In contrast to previous reports using high levels of MYC expression driven by Pdx1-Cre;CAGtTA;TetO-*Myc*[32], two copies of ROSA-driven *Myc* expressed in the pancreas (ROSA[Myc/Myc];Pdx-Cre) alone did not induce tumor formation or affect overall mouse survival (Fig. 3e). However, knock-in of one copy of ROSA-LSL-*Myc* in combination with *Kras*[G12D] (*Kras*[G12D]ROSA[Myc/+];Pdx-Cre) significantly reduced survival compared to KC mice, which was further reduced by knocking in two copies of ROSA-LSL-*Myc* (*Kras*[G12D]ROSA[Myc/Myc];Pdx-Cre) (Fig. 3e). Similar to our findings in human PDA (Fig. 1a, b), CK + SYP + cells occurred within the ductal lesions of KMC mice (Fig. 3f) and co-stained with insulin (Supplementary Fig. 2b). To determine whether the number of ductal SYP + cells change during PanIN progression or whether MYC expression altered their frequency, we quantified the percentage of SYP + cells within ductal structures of KC and KMC mice. While scattered SYP + cells could be detected in KC PanIN lesions, KMC mice had significantly more ductal-associated SYP + cells in equivalent stage PanIN lesions compared to KC mice and the percent of SYP + cells increased as PanIN lesions progressed (Fig. 3g). Taken together, these data indicate a role for MYC in promoting ductal-neuroendocrine plasticity in PDA.

**Ductal-neuroendocrine plasticity and therapeutic resistance.** We analyzed SYP expression in human PDA cell lines, which we have previously shown exhibit elevated levels of MYC expression[29], compared to non-transformed, immortalized pancreas ductal epithelial cells (hTERT-HPNE)[33]. These studies revealed that *SYP* mRNA (Fig. 4a − left) and SYP protein levels (Fig. 4a − right) were highest in the poor-outcome squamous/quasi-mesenchymal subtype relative to the classical subtype or hTERT-HPNE cells. In prostate and lung cancer, increased neuroendocrine differentiation was seen in response to therapeutic pressure[11, 13, 34–36] suggesting that ductal-neuroendocrine lineage plasticity in tumor cells may be an important mechanism contributing to therapeutic resistance. Therefore, we examined whether ductal-neuroendocrine lineage plasticity in PDA could contribute to resistance to standard of care gemcitabine, consistent with our observation that PDA patients with > 5% CK +

SYP + cells recurred more rapidly on gemcitabine adjuvant therapy (Fig. 1d). In Panc1 and MiaPaca2 cell lines, which express moderate to high levels of SYP mRNA and protein (Fig. 4a), gemcitabine treatment was ineffective even at high doses, while the viability of Capan1 and HPAFII cells, that express lower levels of SYP (Fig. 4a), was significantly decreased upon gemcitabine treatment (Fig. 4b). However, gemcitabine did not inhibit viability greater than ~60% in any cell line tested. Gemcitabine treatment resulted in increased mRNA expression of the neuroendocrine markers *SYP* and *SNAP25* over time in MiaPaca2, HPAFII, and Capan1 cells (Fig. 4c). Furthermore, the percentage of MiaPaca1 and Capan1 cells expressing SYP protein increased upon treatment with gemcitabine (Fig. 4d), suggesting that chemotherapeutic stress selects for a more resistant, neuroendocrine phenotype in residual cells. Similarly, gemcitabine-treated PDX tumors displayed a significant increase in SYP expression compared to vehicle-treated control tumors (Fig. 4e). RNA sequencing data from HPAFII cells treated with gemcitabine or PBS control supported these data, showing that additional neuroendocrine genes, neuronal genes, and genes involved in vesicle transport and neurotransmitter release were upregulated upon gemcitabine treatment (Fig. 4f). While we did not observe a significant upregulation of *SNAP25* in the gemcitabine treated RNA sequencing data, other SNARE-associated genes were significantly upregulated, including *CPLX3* (Complexin3) and *VAMP1* (Synaptobrevin1) (Fig. 4f). We also analyzed expression of genes previously shown to be associated with gemcitabine resistance[37, 38] (Supplementary Fig. 3a). Notably, the expression of *RRM1*, *RRM2* and *RRM2B*, reported gemcitabine-regulated genes, remained unchanged upon gemcitabine treatment of HPAFII cells. Additionally, while expression of the gemcitabine transporters *SLC29A1* and *SLC29A3* were also unchanged, cytidine deaminase (CDA), an enzyme that inactivates intracellular gemcitabine and is often activated during gemcitabine resistance, was increased 2-fold upon gemcitabine treatment, which could contribute mechanistically to resistance (Supplementary Fig. 3a).

**MYC regulates neuroendocrine genes and gemcitabine response.** Consistent with our observation that MYC contributes to ductal-neuroendocrine lineage plasticity, knockdown of c-MYC in human PDA cell lines significantly reduced expression of the neuroendocrine markers SYP and ChgA in MiaPaca2 and Panc1 cells (Fig. 5a, b). In Capan1 cells, while SYP expression was below the level of detection by Western analysis, MYC knockdown resulted in a significant decrease in ChgA expression (Fig. 5c). Therefore, we next tested whether loss of MYC would decrease gemcitabine-induced neuroendocrine marker expression, and whether this would be associated with an increased sensitivity to gemcitabine. As expected, exposing human PDA cell lines to gemcitabine increased SYP protein expression and MYC knockdown reduced SYP expression (Fig. 5d). In MiaPaca2 and Capan1 cells, the increase in SYP expression with gemcitabine was attenuated by knocking down MYC, while in Panc1 cells,

**Fig. 3** MYC facilitates ductal-neuroendocrine lineage plasticity in PDA. **a** KPC or KPC *Myc*[fl/+] pancreatic tissues from mice with end stage disease were stained for SYP and Pan-CK by IF. Scale bars indicate 100 μM. Examples of co-stained cells are marked with white arrows. **b** The percentage of CK-SYP dual positive cells within ductal structures relative to the total number of PanCK positive cells was quantified. n = 4 mice per group. Error bars indicate standard error of mean. *P = 0.029 one-tailed Mann Whitney test. **c** Interrogation of MYC ChIP-seq data from KPC tumor cells. GSEA analysis shows enrichment in MYC DNA binding peaks within 1000 bp of genes that are upregulated in NEPC vs. PCA[11]. **d** Schematic depicting generation of the LSL-*Kras*[G12D];LSL-ROSA-*Myc*;Pdx1-*Cre* (KMC) mice. **e** Expression of deregulated MYC in combination with oncogenic KRAS in KMC mice reduces overall survival compared to KC mice. The number of mice and median survival of each arm is shown in the legend. **f** KMC pancreatic tissues were stained for PanCK and SYP by IF. Examples of CK-SYP co-stained cells are indicated with white arrows. Scale bars indicate 100 μM. **g** *Kras*[G12D];Pdx1-Cre (KC) and KMC pancreatic tissues were stained for SYP by IHC. The percentage of SYP positive cells within the ductal epithelium relative to the total number of ductal cells was quantified in 5–10 of each lesion grade per genotype. Error bars indicate standard error of mean. ***P < 0.001, two-way ANOVA

although SYP expression was lower with siMYC, it was still mildly increased by gemcitabine treatment. Importantly, siMYC significantly enhanced the sensitivity of all three PDA cell lines to gemcitabine, correlating with decreased SYP expression with siMYC compared to non-targeting control siRNA (Fig. 5e). Together, these data indicate that MYC contributes to ductal-neuroendocrine lineage plasticity of PDA, which supports chemoresistance.

## Discussion

Accumulating evidence from multiple cancer types indicates that tumor cells display a high degree of cellular plasticity, presumably enabling them to survive in harsh environments. Cellular plasticity drives intratumoral phenotypic heterogeneity, which has recently been linked to therapeutic resistance and disease recurrence[2–4]. While the role of intratumoral phenotypic heterogeneity in tumor biology and therapeutic response is becoming

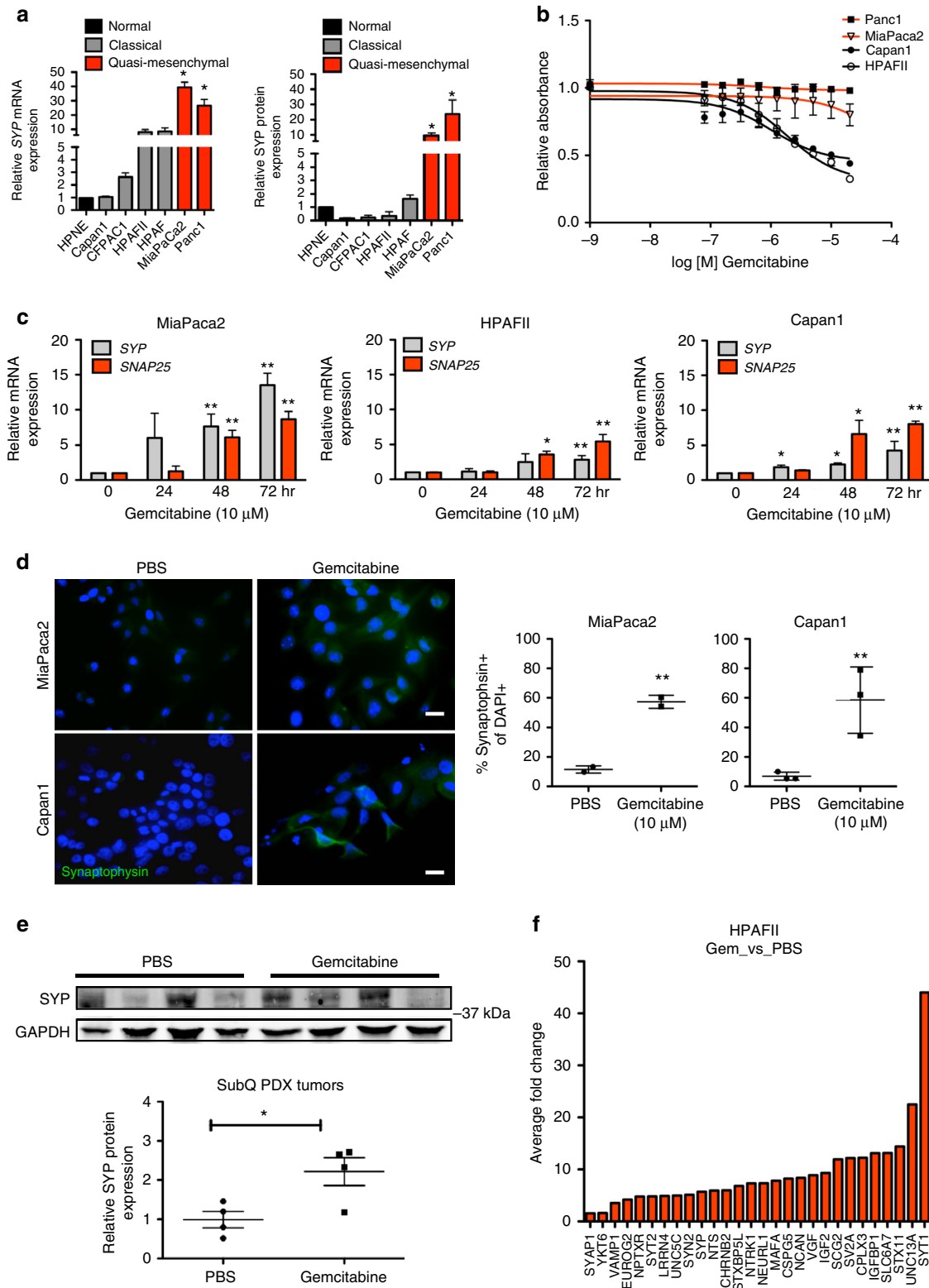

increasingly recognized, there is an urgent need to identify resistance phenotypes in these heterogeneous cancers and the molecular mechanisms underlying the emergence of these phenotypes.

Here, we show that a subset of human PDA tumors display neuroendocrine differentiation features, where ductal tumor cells express neuroendocrine markers, as has been described in prostate and lung cancers[9, 11, 13, 27, 36]. Higher percentages of tumors cells co-expressing CK and SYP correlated with shortened disease-free survival in patients with PDA. Further, NEPC-associated genes are enriched in PDA tumors of the poor outcome squamous subtype and SYP expression is highest in the squamous/quasi-mesenchymal subtype of human PDA cell lines. Consistent with these data, we also detected CK-SYP dual positive CTCs from patients with PDA and find that individual CTCs from KPC mice upregulate the NEPC gene signature more frequently than matched tumor single cells, suggesting two possibilities: (1) that CK + SYP + cells are more migratory/invasive and possess the inherent ability to extravasate into the bloodstream or (2) that once in the bloodstream, CTCs expressing neuroendocrine markers are more likely to survive in the harsh environment. Further studies will be needed to distinguish between these two possibilities.

We also report here that tumor cells capable of expressing SYP survive passaging through mice in PDX models, suggesting that these dual positive cells are tumor-derived, rather than hyperplasia of adjacent normal islet cells. While these dual lineage positive cells may represent tumor cell phenotypic heterogeneity that can expand or contract due to selective pressure, we confirmed that these cells can be derived from the exocrine compartment through lineage/differentiation-state plasticity using in vivo lineage tracing in KCY mice[26] with YFP induction in adult acinar cells. In these studies, we detected YFP positive cells that co-express SYP in ductal lesions, demonstrating that these dual positive cells do not originate from the endocrine (islet) compartment. It is also worth noting that in these studies we observed single YFP + SYP + cells within the tumor stroma (Fig. 2b). This data is consistent with other studies demonstrating YFP + cells in the stroma in KCY mice[26] and further supports our data shown here that circulating tumor cells are frequently CK + SYP + . We further observed that genes involved in EMT are upregulated upon gemcitabine treatment (Supplementary Fig. 3b). We also found evidence of SYP + cells that co-stained with the stem marker Nestin in human PDA samples (Supplementary Fig. 3c), therefore it is interesting to speculate that the plasticity of these neuroendocrine marker-positive cells is associated with stem/progenitor capacity and future experiments are planned to more specifically address this possibility. Indeed, endocrine and exocrine lineages arise from a common bi-potential progenitor in pancreas development[39] and multiple studies have demonstrated the ability of adult exocrine cells in the normal pancreas to transdifferentiate into insulin-producing Beta (β) cells[40, 41]. While scattered neuroendocrine cells in human PDA has been reported[10, 42], the degree of ductal-neuroendocrine lineage heterogeneity in PDA and its potential role in therapeutic resistance has not been previously described. We view the presence of these EMT and stem/progenitor molecular features in PDA cells with neuroendocrine differentiation as representing an underlying plasticity that supports their aggressive behavior and therapeutic resistance.

Several published studies have demonstrated a necessary role for N-MYC and c-MYC in regulating neuroendocrine transdifferentiation of prostate adenocarcinoma resulting in the formation of the aggressive neuroendocrine-differentiated subtype[11, 12, 14, 27]. Similarly, we demonstrate here that c-MYC helps to regulate ductal-neuroendocrine plasticity of PDA, where we show that c-MYC bound genes in PDA are enriched in genes specifically upregulated in neuroendocrine-differentiated prostate cancer and that loss of c-MYC in mouse and human PDA cells decreases neuroendocrine marker expression. To further study MYC's potential role in ductal-neuroendocrine lineage plasticity, we utilized our KMC mice, which express pancreas-specific suboncogenic levels of MYC in combination with mutant KRAS. In these mice, low-level MYC expression alone is insufficient to drive tumor formation, however, deregulated MYC expression accelerated mutant KRAS-induced lesion formation. We found that the percentage of ductal-associated SYP positive cells is increased upon expression of MYC in KMC vs. KRAS only (KC) mice at matched PanIN stage. We also observed an increase in this population with PanIN progression, further supporting a role for this process in the development of aggressive disease.

Interestingly, we also found that neuroendocrine marker expression increased in human PDA cell lines or PDX tumors exposed to standard of care chemotherapy, suggesting that ductal-neuroendocrine lineage plasticity may function as a mechanism of therapeutic resistance in PDA. This finding could help explain the rapid recurrence of patients showing over 5% ductal-neuroendocrine lineage heterogeneity treated with adjuvant gemcitabine. Mechanistically, knockdown of MYC decreased the expression of neuroendocrine markers and consistent with this, reducing MYC increased sensitivity to chemotherapy. Thus, targeting MYC to decrease ductal-neuroendocrine lineage plasticity in PDA cells may increase tumor cell response to standard-of-care gemcitabine. While clinically targeting MYC remains an elusive goal, progress is being made to inactivate MYC through inhibition of BET Bromo-domain containing proteins[43, 44] and through PP2A activation leading to MYC S62 dephosphorylation and protein destabilization[45, 46].

In conclusion, we show that ductal-neuroendocrine lineage heterogeneity in PDA contributes to poor outcome and therapeutic resistance, demonstrating a role for lineage plasticity in aggressive PDA. The MYC oncoprotein contributes to this cell

**Fig. 4** Ductal-neuroendocrine plasticity contributes to gemcitabine resistance. **a** *SYP* mRNA (left panel) and SYP protein (right panel) expression in human PDA cell lines. Fold change in *SYP* mRNA expression was determined by qRT-PCR (normalized to *TBP*) relative to the immortalized duct epithelial cell line hTERT-HPNE (HPNE). Fold change in SYP protein expression (normalized to GAPDH) relative to HPNE was determined by Western blot. *P* values indicate significance when comparing to HPNE. Results representative of $n = 3$ experimental replicates. **b** Cells were treated with the indicated doses of gemcitabine and measured using MTS assay. Results represent average of $n = 3$ experimental replicates. **c** Cells were treated with gemcitabine for the indicated time points and neuroendocrine genes were assessed by qRT PCR (normalized to *TBP*). *P* values indicate significance when comparing each single gene to the zero hour time point. Results representative of $n = 3$ experimental replicates. **d** MiaPaca2 and Capan1 cells were treated with gemcitabine for 72 h, fixed and stained for SYP by IF. The percentage of SYP positive cells of total DAPI positive cells was quantified for triplicate wells, 3 images/well. Representative images and quantitation from three experimental replicates are shown. Scale bars indicate 20 μM. **e** Western analysis of tumor lysates from subcutaneous (SubQ) PDX tumors treated with vehicle or gemcitabine. SYP expression relative to GAPDH was quantified for each tumor. **f** HPAFII cells were treated with PBS or gemcitabine ($n = 3$ experimental replicates) and RNA sequencing was performed. Fold change in neuroendocrine genes, neuronal genes, and genes involved in vesicle trafficking for gemcitabine treated vs. PBS cells are shown. For all panels, error bars indicate standard deviation and *$P < 0.05$, **$P < 0.01$ Student's two-tailed t-test

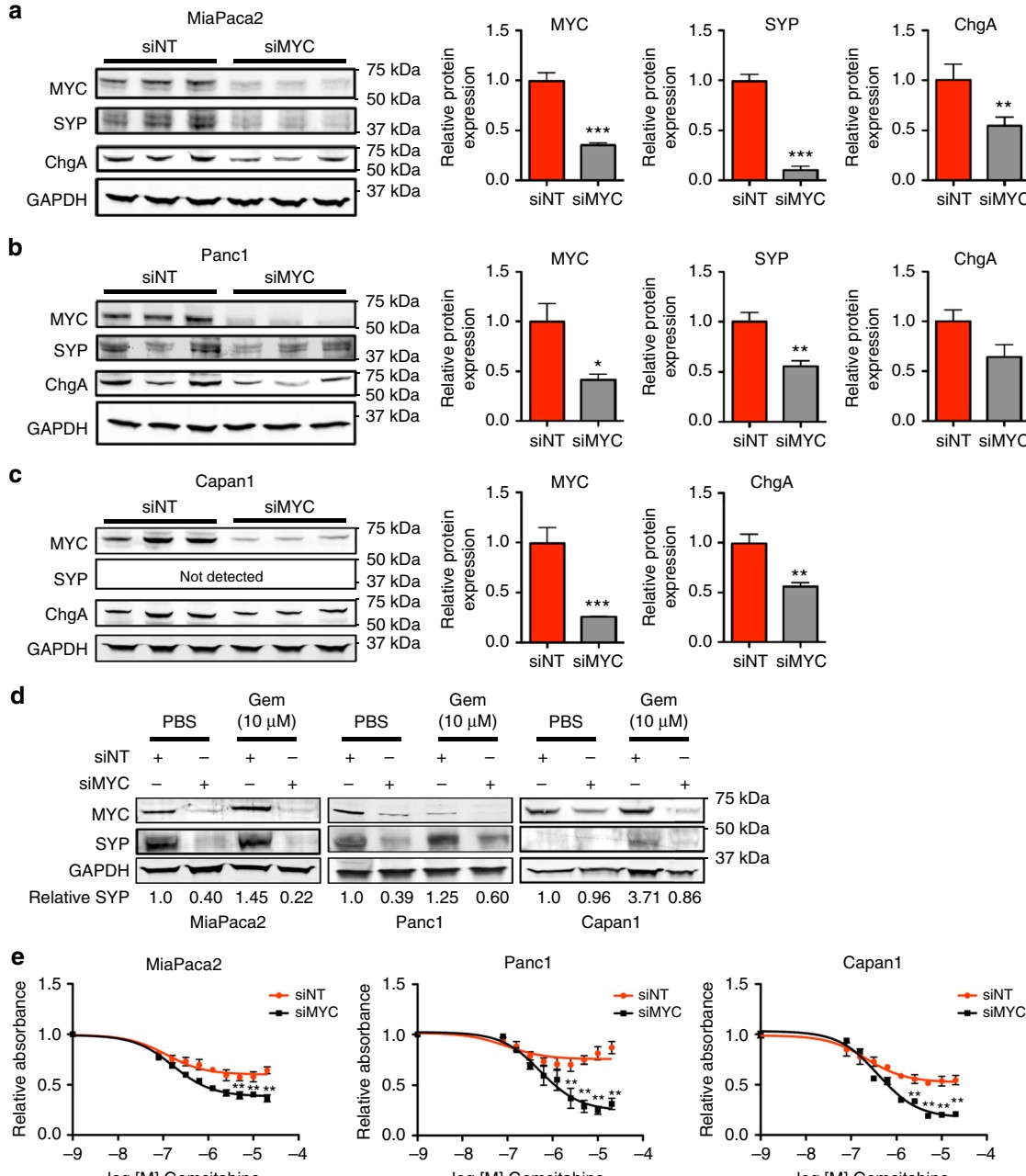

**Fig. 5** MYC regulates neuroendocrine markers and gemcitabine response. **a–c** Loss of MYC in human PDA cell lines decreases neuroendocrine marker expression. Cells were transfected with siRNA targeting c-MYC or non-targeting control siRNA (siNT), total protein was harvested after 72 h and assessed by Western blot. Fold change in protein expression (relative to GAPDH) in siMYC cells normalized to siNT is shown for three independent experimental replicates. Error bars indicate standard deviation. *$P < 0.05$, **$P < 0.01$, ***$P < 0.001$, Student's two-tailed t-test. **d** Cells were treated with siNT or siMYC for 24 h, then treated with PBS or gemcitabine for 72 h, and assessed by Western blot. SYP protein expression (normalized to GAPDH) is shown relative to siNT PBS control. gem = gemcitabine. Error bars indicate standard deviation. **e** Cells were treated with siMYC or siNT for 24 h, and then treated with increasing doses of gemcitabine for 72 h. MTS assay was performed. siNT and siMYC curves are shown relative to PBS control for $n = 3$ experimental replicates. Error bar indicate standard deviation. *$P < 0.05$, **$P < 0.01$, two-way ANOVA

lineage heterogeneity in PDA and combined oncogenic KRAS and activated MYC could drive aggressive phenotypic heterogeneity that may allow the emergence of a chemotherapy resistant state enriched for neuroendocrine differentiation in PDA patients. Indeed, the poor outcome squamous subtype of PDA is reported to have high MYC pathway activity[24] and the neuroendocrine gene signature is enriched in this subtype. While difficult to obtain, tumor samples collected while patients are progressing on chemotherapy would help reveal such a mechanism.

## Methods

**Cell culture and siRNA transfection**. Pancreatic cancer cells used in these studies were obtained from Michel Ouellette (University of Nebraska Medical Center, Omaha, NE) and from Joe Gray (Oregon Health and Science University, Portland, OR) and maintained in DMEM with 10% FBS. Cell lines were tested for mycoplasma contamination, identities were confirmed by STR profiling and no cell lines used were found in the database of misidentified cell lines. Transient MYC knockdown was performed using siRNAs (25 nM) (Dharmacon, Pittsburgh, PA) and DharmaFECT I transfection reagent according to DharmaFECT protocols. Non-targeting siRNA (D-001206-14) was used as a control. Cells were incubated with siRNAs for 24 h and then cells were seeded for additional experiments. For MTS assays, cells were trypsinized, counted and 5,000 cells were plated in 96-well

plates and allowed to adhere overnight. Cells were treated with PBS or increasing doses of gemcitabine (ranging from 390 nM to 100 μM) for 72 h and analyzed using Cell Titer 96 Aqueous One Solution/MTS (#G3580, Promega, Madison, Wisconsin) on the GloMax Microplate reader (Promega, Madison, Wisconsin).

For IF analysis of Synaptophysin expression after gemcitabine treatment in pancreatic cancer cells, cells were seeded in 96-well plates, allowed to adhere overnight, then treated with 10 μM gemcitabine × 72 h. Images were taken using the Nikon Spinning Disk. The percent of Synaptophysin positive cells of total DAPI-positive cells were quantitated, 3 images/well in triplicates. For all other experiments in which cells were treated with a single dose of gemcitabine, 10 μM gemcitabine was used.

**Genetically engineered mouse models.** All animal studies were conducted in compliance with ethical regulations and were approved by OHSU IACUC (protocol number ISO00003989). The generation and characterization of the $Kras^{G12D}p53^{R172H/+}$Pdx-Cre[47], KPC and KPC $Myc^{fl/+}$[28] and ROSA-LSL-Myc[30] mice have been previously described. The LSL-$Kras^{G12D}$;Pdx1-Cre mice[31] were obtained from Dr. Kerry Campbell at Fox Chase Cancer Center. All mice were a mixed B6/129 background, and both males and females were used. KC and KMC pancreatic tissue were stained for Synaptophysin by IHC. The percentage of Synaptophysin-positive cells within the ductal epithelium relative to the total number of ductal cells was quantified in 5–10 of each lesion grade per genotype.

For in vivo lineage tracing, $Kras^{G12D}$;Ptf1a-CreERT;ROSA-YFP (KCY) mice[26] were treated with 5 mg tamoxifen once a day for 5 consecutive days to induce acinar cell specific recombination. Mice were then treated with 250 μg/kg caerulein once a day for five days to accelerate acinar cell transformation and allowed to recover for 9 weeks. $n = 10$.

**Patient-derived xenografts.** 2 mm³ pieces of primary pancreatic tumor tissue, resected from patients with informed written patient consent (collected in collaboration with the Brenden-Colson Center for Pancreatic Care, IRB approved, IRB00003330), were coated in Matrigel and implanted subcutaneously into the flank of 6-week-old NOD.Cg-*Prkdcscid Il2rgtm1Wjl*/SzJ (NSG) mice (passage 1). Once tumors reached 2 cm in diameter, animals were sacrificed and a 2 mm³ piece of the passage 1 tumor was then implanted as above into a second NSG mouse (passage 2). Tumors were harvested at 2 cm in diameter and formalin fixed paraffin embedded for further analysis. For experiments with gemcitabine treatment, an N1 pancreatic adenocarcinoma PDX tumor was mechanically dissociated and grown in cell culture for > 7 passages. $1 \times 10^6$ cells were injected subcutaneously into the mouse flank. Mice were randomized into arms and when average tumor volume for each group reached 150 mm3, tumors were treated with Vehicle or gemcitabine (100 mg/kg I.P). All mice were sacrificed and tumors harvested when the first mouse became moribund.

**Human primary tumor analysis.** For survival analysis, human primary tumors from resected pancreatic ductal adenocarcinoma (from patients with informed, written consent) were selected from our tumor bank (Brenden-Colson Center for Pancreatic Care, IRB approved, IRB00003330). Patients were well matched for age, gender, AJCC stage, histologic grade, perineural invasion, lymphovascular space invasion, R0 resection status, and adjuvant treatment (Supplementary Table 1). Patients who received neoadjuvant therapy, had M1 disease after final pathology, or who died without evidence of recurrence were excluded. Disease-free survival was measured from date of resection to earliest evidence of disease recurrence. Based on nuclear morphology and Cytokeratin staining, the total percentage of tumor epithelium co-staining for Cytokeratin and Synaptophysin across entire sections were graded as < 5% or ≥ 5%. This distinction was selected based on the fact that co-staining cells are seen at low frequencies in normal duct epithelium. Structures resembling islets were not included. A second cohort of patients with tumors representing various grades was similarly stained and quantified for CK-SYP co-staining ($n = 10$).

**Human CTC capture and analysis.** Five treatment-naive patients with biopsy-proven pancreatic ductal adenocarcinoma were enrolled for analysis after informed consent was obtained (UM IRB HUM00025339). Blood was collected and analyzed, as previously described[48]. Briefly, peripheral venous blood was obtained through venipuncture and collected into Cell Save preservative tubes (Janssen Pharmaceuticals, Raritan, New Jersey). One milliliter was applied via microfluidic pump to a geometrically enhanced differential immunocapture (GEDI) chip functionalized previously with capture antibodies specific to epithelial cell adhesion molecule (EpCAM). After application of blood, the chip was washed with running buffer and antibodies to Cytokeratin and Synaptophysin conjugated to fluorophores and DAPI were applied to stain cells captured on the GEDI chip, as previously described. Stained cells were counted blindly as previously described. Scale bars indicate 100 microns.

**Mouse CTC dataset analysis.** First, we obtained a list of 585 upregulated and 549 downregulated genes in NEPC vs. PCA from the Beltran et al.[11] and checked for expression of these genes in the TuGMP (black bars) and matched GMP circulating tumor cells (CTCs–red bars) from the single cell mouse RNA-Seq data from Ting

et al.[23], accession number: GSE51372. For each single cell RNAseq data, we identified expressed genes by applying a tag threshold of 100 and then calculated a ratio of the number or these genes that were in the up vs. down in NEPC vs. PCA.

**Immunofluorescence.** Serial paraffin sections were de-paraffinized, rehydrated, and blocked in 3% BSA. They were then incubated with primary antibodies overnight at 4 °C. Antibodies used were: CK8/18 (1:200, Fitzgerald, #20R-CP004, Suffolk, England), Synaptophysin (1:100, Sigma/Cell Marque #336R-95/clone MRQ-40, St. Louis, MO), Insulin (1:100, Cell Signaling Technology, #4590 S, Danvers, MA), Pan-Cytokeratin AE1/AE3 (1:200, Santa Cruz, #sc-81714, Dallas, TX), Nestin (1:100, Abcam #22035, Cambridge, UK). Sections were then incubated with AlexaFluor 594 or 488 secondary antibodies (1:200, Life Technologies, Grand Island, NY) and, DAPI (1:2000) and mounted using prolong gold anti-fade reagent (Life Technologies, Grand Island, NY).

**Immunohistochemistry.** Unstained sections were de-paraffinized, rehydrated, and placed in Target Retrieval Solution (Dako, Carpinteria, CA) for 10 min at 100 °C. After cooling for 20 min, slides were quenched with 3% hydrogen peroxide for 5 min, followed by blocking with 3% goat serum and primary antibody incubation (Synaptophysin, 1:200, Sigma/Cell Marque #336R-95/clone MRQ-40, St. Louis, MO) overnight at 4 °C. Sections were then incubated with anti-biotin secondary antibodies (1:1000) and labeling was then detected with the Vectastain ABC kit (Vector Laboratories). Slides were mounted using Vectamount mounting media (Vector Laboratories).

**RNA sequencing and gene set enrichment analysis .** HPAFII cells were treated for 72 h with 10 μM gemcitabine, RNA was isolated using RNeasy mini-kit (Qiagen) according to the manufacturer's protocol. RNA-seq libraries were generated using standard TruSeq reagents for analysis in the MPSSR at OHSU using an Illumina HiSeq2500. In total 50 base pair single reads sequences were generated and called using the standard Illumina pipeline, requiring at least 20 M reads per library. Sequence reads were trimmed to 44 bases and aligned with the Bowtie short read aligner version 1.0.0. We allowed 3 miss-matches, but required unique best matches. Gene summaries were calculated based on UCSC RefSeq gene annotation for hg19 and used for statistical analysis with DESeq2. Reads Per Kilo-base per Million (RPKM) were calculated for use in gene set enrichment analysis (GSEA) using the following web site: http://www.broad.mit.edu/gsea/.

For GSEA analysis to determine if NEPC upregulated genes were enriched in human PDA subtypes, we used the 96 PDAC ICGC RNAseq dataset from Bailey et al.[24] and their designated subtypes for each sample identified, looking for enrichment of the upregulated genes in the NEPC vs. PCA comparison from Beltran et al.[11]. We used GSEA software comparing each individual subtype to the rest.

For analysis of MYC chIP seq data in KPC tumor cells from Walz et al.[28], we selected peaks with an FDR q-value of less than or equal to 0.01, annotated them with the 5 prime ends of Ref-Seq genes and then selected all genes with one or more MYC ChIP peaks within 1000 bases of their 5 prime end. Next, we created a ranked list of genes from the Beltran NEPC vs. PCA comparison and used GSEA software to test for enrichment of the MYC ChIP genes in the NEPC vs. PCA ranked list.

**Quantitative RT-PCR.** RNA was isolated using RNeasy mini-kit (Qiagen) according to the manufacturer's protocol. cDNA was generated using the High Capacity cDNA Reverse Transcription kit (ABI, Foster City, California). qRT-PCR analysis was performed using SYBR Green reagent (Invitrogen) for Synaptophysin (*SYP*) and *SNAP25* normalized to the internal control TATA-box binding protein (*TBP*). Primers were validated by performing a standard melt curve analysis. Primer sequences are listed in Supplementary Table 3. For qRT-PCR, the cycle numbers for each run were used to calculate fold change using the ddt(ct) method between each sample and the average of the control sample (set as 1) in gene expression as follows:

dt(ct) = SYP cycle # - GAPDH cycle #
ddt(ct) = dt(ct) − average dt(ct) of normal samples
Fold Change = Power(2,ddt(ct))

**Western blot analysis.** Cells were homogenized in ice-cold AB-MYC lysis buffer (1 M Tris pH 7.5, 5 M NaCl, 10% Triton X-100, 10% DOC, 20% SDS, 0.5 M EDTA, 1× protease and phosphatase inhibitors (Roche, Indianapolis, Indiana)), passed through a 30 gauge needle, and cleared by centrifugation at 4 °C, 13,000×$g$ for 10 min. Protein concentration was determined using the DC protein assay (BioRad, Hercules, California). Proteins were separated by SDS–PAGE and transferred to nitrocellulose membranes. Membranes were blocked in blocking buffer (Aqua block, 1:1 in PBS, East Coast Bio, North Berwick, Maine) for 1 h and incubated in primary antibody in blocking buffer overnight. Membranes were then washed with PBS-T (Phosphate-buffered saline, 0.1% Tween-20), incubated in Fluor 680 and IRDye 800 secondary antibodies (1:10,000) for 1 h, washed with PBS-T and visualized using the Odyssey IR imager (LI-COR, Lincoln, Nebraska) that can detect both Fluor 680 and IRDye 800 secondary antibodies. Quantification of Western blots was done using the Odyssey IR software, version 1.2 (LI-COR). The

following primary antibodies were used: Synaptophysin (1:500, Sigma/Cell Marque #336R-95/clone MRQ-40, St. Louis, MO)), MYC (1:1000, Abcam #32072, Cambridge, UK), Chromogranin A (1:100, Abcam #15160, Cambridge, UK) and GAPDH (1:10,000, Invitrogen #AM4300, Carlsbad, CA). Images of uncropped Western blots can be found in Supplementary Fig. 4.

**Data and statistics**. Three or more independent biological replicate experiments were performed in all cases, except when primary patient materials were used. All standard errors were calculated from biological replicates. Unless otherwise stated in the figure legends, $P$ values were calculated using a standard Student's $t$ test analysis (two-tailed distribution and two-sample unequal variance) to determine statistical significance as indicated in the graphs. $P$-values for relevant comparisons are given. If no $P$ value is shown, the comparison is not relevant or not significant. To determine if CK-SYP staining is an independent risk factor for poor outcome in PDA patients, we performed multivariate analysis to determine if clinical stage, grade or age significantly altered relationship between CK-SYP co-staining and days-to-recurrence. Briefly, we used R software to perform multiple linear regression, fitting the following models: (1) DaysToRecurrence ~ CoStaining (2) DaysToRecurrence ~ CoStaining + ClinicalStage (3) DaysToRecurrence ~ CoStaining + Grade (4) DaysToRecurrence ~ CoStaining + Age. These potential confounding variables altered the magnitude of the co-staining effect by less than 10%, therefore we determined the effect of co-staining to be independent of these potential confounders.

**Data availability**. The authors declare that all data supporting the findings of this study are available within the paper (and its Supplementary Information files) or will be made available upon request. Accession number for RNA-sequencing of HPAFII cells treated with PBS or gemcitabine from this manuscript: GSE106336. Accession number for single-cell mouse RNA-Seq data from Ting et al.[23]: GSE51372.

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

## Acknowledgements

Primary pancreatic tumor tissue, resected from patients with informed written patient consent, was supplied by the Brenden-Colson Center for Pancreatic Care (BCCPC). H&E staining was performed by the OHSU Histopathology Shared Resource, and short read sequencing assays (RNA-seq) were performed by the OHSU Massively Parallel Sequencing Shared Resource. Multivariate statistical analysis was performed with assistance from the OHSU Biostatistics Shared Resource. These OHSU Shared Resources are supported by the Knight NCI Cancer Center Support Grant 5P30CA069533. We thank Dr. Joe Gray and Dr. Michel Ouellette for providing cell lines. ASF was supported by a Collins Medical Trust Fund Award, Knight Cancer Institute Career Development Award, and R01 CA196228. RCS was supported by R01s CA196228, CA100855 and CA129040, DOD BC061306, and the Anna Fuller and Brenden-Colson Foundations. B.L.A-P. was supported by NRSA1F32CA192769-01. M.M.J. was supported by NRSA1F32CA213764-01.

## Author contributions

A.S.F., M.M.J., B.L.A-P. and R.C.S. designed experiments and wrote the paper. S.L.M., E. D.P., and A.D.R. performed human CTC analysis. H.C.C. and M.T.H. performed KCY analysis. A.S.F. generated and characterized the KMC mouse model. C.L. and D.S. performed histopathological analysis of KMC mice. B.L.A-P. and M.C.T. generated human PDX tumors and performed analyses. L.M.H. performed bioinformatics analysis of NEPC gene expression in PDA subtypes. C.P. performed all additional bioinformatics analysis. J.P.M., O.J.S., N.M., D.J.M., and B.L.A.-P. performed KPC $Myc^{fl/+}$ analysis. T.S. A. performed SYP IF on fixed cells and quantification of SYP staining. P.J.W., J.L., N.D. K., and N.E. performed IF analysis of human PDA samples. P.J.W. and J.L. performed patient survival analysis. M.M.J. performed and analyzed RNA sequencing, performed all Western and qRT PCR analysis and MYC knockdown studies in PDA cells. Z.P.J. performed additional experiments. All authors contributed to data analysis and read the manuscript. R.C.S. and B.C.S. supervised the project.

## Additional information

**Competing interests:** The authors declare no competing financial interests.

