## [Peer Review File · Nature Communications]

Reviewers' comments:

Reviewer #1 (Remarks to the Author):

Farrel and colleagues show evidence of a new role of MYC in promoting ductal-neuroendocrine lineage plasticity in pancreatic ductal adenocarcinoma. The manuscript is nicely written and the results are presented in a clear and sensible manner. However, some points need to be addressed before publication.

Comments:

- The abstract reads:

"Mechanistically, MYC binding was enriched at neuroendocrine lineage genes in mouse tumor cells and loss of MYC reduced ductal-neuroendocrine lineage heterogeneity, while MYC expression in KRAS mutant mice increased this phenotype." As far as I understand it, in KRas only-driven tumors there is Myc expression, but it would not be deregulated as is in KMC mice. I would recommend then changing the phrase to read: "...while MYC deregulated expression in KRAS mutant mice increased this phenotype".

- Figure 4a: Panc1 cells are the only cells shown here as having high levels of SYP mRNA and compared with two cell lines having low levels of SYP. Extrapolating conclusions regarding the general behavior of SYP-high vs. SYP-low in response to gemcitabine treatment with n=1 is quite risky.

- Figure 5

1) MiaPaca2 and Panc1 cells are studied in 5a, 5b, and 5c. But then Capan1 are added in 5d. What happens to Myc, ChgA and SYP in Capan1 cells in response to Myc knockdown and/or Gem treatment? Why aren't they shown?

2) For figure 5c it is concluded that "as expected...this increase in SYP expression was attenuated by knocking down Myc". Actually, in Panc1 cells, the gemcitabine-induced increase in SYP is from 1 to 1.25 (control cells) whereas after Myc knockdown it is 0.39 to 0.60, hence there is a 50% increase in SYP caused by gemcitabine even in the absence of Myc (or, in other words, the relative increase is 0.21 after Myc knockdown compared to 0.25 in control cells - Myc knockdown here has no effect on the gemcitabine-induced increase in SYP). Again, the comparison with Capan1 as well would have been helpful, and the same conclusion cannot be drawn from these 2 cell lines as shown. Curiously, while gemcitabine massively reduces Myc levels, the SYP levels are not reduced in the way they are after Myc knockdown.

3) In the legend to figure 5d, it says that the cells were treated plus or minus gemcitabine when in fact they are treated with different doses. No 'minus gemcitabine' cells are shown, unless perhaps the viability is relative to such cells? (It is stated that the viability is relative to siNT, but the siNT viability varies notably so it's not clear what the control actually is).

4) What is the concentration of gemcitabine used in figure 5c?

5) Important: It is notable that gemcitabine actually causes a decrease in Myc levels, but only in Panc1 cells, not in MiaPaca2 (again, one wonders what happens in Capan1 cells). It is impressive therefore that the slight extra suppression of Myc levels after knockdown in Panc1 cells (hardly visible by the western) has such a large effect on SYP levels and on gemcitabine-reduced viability.

6) Major point: In 5b, the Myc and ChgA blots are extremely similar. Please, double check. In Figure 5a, the levels of Myc and ChgA do not mirror each other precisely like this.

Suggestion: There are a large number of acronyms and abbreviations that make the paper quite hard to read. Consider replacing them where possible and use the full term, for example: squamous/QM (there are only 5 mentions of QM, so it seems unnecessary to abbreviate it) ductal-NE (while NE is commonly used, since 'ductal' is spelled out, is it necessary to abbreviate NE? It would be easier to read throughout by talking about a "neuroendocrine phenotype" too.)

Reviewer #2 (Remarks to the Author):

In this interesting and provocative manuscript by Dr. Sears and colleagues, the authors propose that deregulated c-Myc promotes a NE differentiation program stochastically during PDAC development, leading to worsened disease outcomes in patients and gemcitabine resistance. This is an addition to the field, and opens new avenues for consideration. Consideration to a few points will strengthen the manuscript for the field.

1. The NE phenotype – is it an association or is it causal in the features described here? For example, both EMT and Sox2 expression are discussed as occurring in the same PanIN/PDAC cells, but the authors are not claiming that EMT or Sox2 expression are driving the NE phenotype. Can the authors please clarify this topic in their manuscript.
2. Gemcitabine resistance – is this a pharmacological or pharmacodynamic resistance mechanism in the cells/models used in this paper?
3. Myc knockdown – this is known to be lethal in a cell autonomous fashion so the last experiments with this in human PDAC cells are not very informative. The alternative, of suppressing one copy of Myc in vivo using mice (as Owen Sansom has done in the colon) would be informative regarding the percentage of NE cells in PanIN/PDAC, as this is not lethal.
4. Is CK-SYP expression an independent risk factor for poor outcomes in PDAC patients, by a multi-variate analysis? Possibly, the small sample size used here will be large enough to assess this point; and in that case I suggest the authors collaborate with those who have large tumor banks and outcomes data (Andrew Biankin, for example).

Response to Referees' comments:

We would like to take this opportunity to thank the reviewers for their time and effort in reviewing our manuscript entitled "MYC regulates ductal-neuroendocrine lineage plasticity in pancreatic ductal adenocarcinoma associated with poor outcome and chemoresistance". We have addressed all of the concerns as detailed below and feel that with these changes our manuscript is stronger and now suitable for publication in *Nature Communications*.

Reviewers' comments:

Reviewer #1 (Remarks to the Author):

Farrel and colleagues show evidence of a new role of MYC in promoting ductal-neuroendocrine lineage plasticity in pancreatic ductal adenocarcinoma. The manuscript is nicely written and the results are presented in a clear and sensible manner. However, some points need to be addressed before publication.

Comments:

- The abstract reads:

"Mechanistically, MYC binding was enriched at neuroendocrine lineage genes in mouse tumor cells and loss of MYC reduced ductal-neuroendocrine lineage heterogeneity, while MYC expression in KRAS mutant mice increased this phenotype." As far as I understand it, in KRas only-driven tumors there is Myc expression, but it would not be deregulated as is in KMC mice. I would recommend then changing the phrase to read: "...while MYC deregulated expression in KRAS mutant mice increased this phenotype".

We thank the reviewer for catching this error and we agree. We have therefore changed the wording in the abstract as suggested.

- Figure 4a: Panc1 cells are the only cells shown here as having high levels of SYP mRNA and compared with two cell lines having low levels of SYP. Extrapolating conclusions regarding the general behavior of SYP-high vs. SYP-low in response to gemcitabine treatment with n=1 is quite risky.

We agree and have now added data showing MiaPaca2's response to gemcitabine. These cells express moderate to high levels of SYP, and their response falls in between Panc1 and Capan1/HPAFII

(low SYP) in the spectrum of gemcitabine response (**see new Figure 4b**). With the inclusion of this additional data, we believe we have strengthened our original observation that PDA cell lines with higher levels of SYP expression (Panc1 and MiaPaca2) respond poorly to gemcitabine.

- Figure 5

1) MiaPaca2 and Panc1 cells are studied in 5a, 5b, and 5c. But then Capan1 are added in 5d. What happens to Myc, ChgA and SYP in Capan1 cells in response to Myc knockdown and/or Gem treatment? Why aren't they shown?

*We originally chose to use Panc1 and MiaPaca2 cell lines for our studies assessing neuroendocrine marker expression upon MYC knockdown because these cell lines express moderate-high expression of SYP at baseline, and thus it is easiest to see altered SYP expression with MYC knockdown. However, as the reviewer suggested, we knocked down MYC in Capan1 cells and assessed SYP and ChgA expression. While in these studies, SYP protein was below the level of detection by Western analysis, ChgA was detectable in siNT controls, and was significantly reduced upon MYC knockdown (**new Figure 5c**).*

*Additionally, we have now included Capan1 cells in our analysis of MYC knockdown and gemcitabine treatment. While SYP was lowly expressed (almost below the level of detection by Western analysis) in both the siNT and siMYC conditions, we observed induction of SYP expression upon gemcitabine treatment, which was reduced upon MYC knockdown in gemcitabine-treated cells (**new Figure 5d**).*

2) For figure 5c it is concluded that "as expected...this increase in SYP expression was attenuated by knocking down Myc". Actually, in Panc1 cells, the gemcitabine-induced increase in SYP is from 1 to 1.25 (control cells) whereas after Myc knockdown it is 0.39 to 0.60, hence there is a 50% increase in SYP caused by gemcitabine even in the absence of Myc (or, in other words, the relative increase is 0.21 after Myc knockdown compared to 0.25 in control cells - Myc knockdown here has no effect on the gemcitabine-induced increase in SYP). Again, the comparison with Capan1 as well would have been helpful, and the same conclusion cannot be drawn from these 2 cell lines as shown. Curiously, while gemcitabine massively reduces Myc levels, the SYP levels are not reduced in the way they are after Myc knockdown.

*We agree with the reviewers that Panc1 cells respond differently to gemcitabine than MiaPaca2 in these studies in that while MYC knockdown in Panc1 cells reduces SYP expression, it does not seem to affect SYP increase upon gemcitabine treatment. In order to strengthen our data and resulting conclusions, as suggested, we have now included Capan1 in our analysis. In Capan1 cells, we show that gemcitabine treatment in Capan1 substantially induces SYP expression, which can be prevented with knockdown of MYC (**new Figure 5d**). Importantly (and consistent with results in MiaPaca2 cells), gemcitabine treatment had no effect on MYC levels in Capan1 cells. The effect of gemcitabine on MYC expression in Panc1 cells is interesting although we do not know the cause. We have clarified cell line differences and our conclusions in the text.*

3) In the legend to figure 5d, it says that the cells were treated plus or minus gemcitabine when in fact they are treated with different doses. No 'minus gemcitabine' cells are shown, unless perhaps the viability is relative to such cells? (It is stated that the viability is relative to siNT, but the siNT viability varies notably so it's not clear what the control actually is).

*We thank the reviewers for bringing this figure legend error to our attention. We have now edited the text in the Figure 5 legend (**new Figure 5e**) to clarify that cells were treated with siNT or siMYC and PBS or increasing doses of gemcitabine. Each siNT or siMYC curve was then normalized to the PBS control.*

4) What is the concentration of gemcitabine used in figure 5c?

*The concentration of gemcitabine used in Figure 5C (**new Figure 5d**) was 10 μ M. We have now included this information in the Methods section and the figure itself.*

5) Important: It is notable that gemcitabine actually causes a decrease in Myc levels, but only in Panc1 cells, not in MiaPaca2 (again, one wonders what happens in Capan1 cells). It is impressive therefore that the slight extra suppression of Myc levels after knockdown in Panc1 cells (hardly visible by the western) has such a large effect on SYP levels and on gemcitabine-reduced viability.

*We have now included Capan1 cells in our analysis, showing that gemcitabine treatment in Capan1 robustly induces SYP expression, which can be prevented with knockdown of MYC. In Capan1 cells, gemcitabine treatment had no effect on MYC levels, similar to what we show in MiaPaca2 (**new Figure 5d**). We agree that the reduction in MYC by gemcitabine in Panc1 cells is interesting, although do not know its cause. We do visibly see total loss of MYC with gem + siMYC by Western, corresponding to increased sensitivity to gemcitabine.*

6) Major point: In 5b, the Myc and ChgA blots are extremely similar. Please, double check. In Figure 5a, the levels of Myc and ChgA do not mirror each other precisely like this.

We are thankful to the reviewer for bringing this mistake to our attention. We have replaced the MYC blot with the correct one.

Suggestion: There are a large number of acronyms and abbreviations that make the paper quite hard to read. Consider replacing them where possible and use the full term, for example:

squamous/QM (there are only 5 mentions of QM, so it seems unnecessary to abbreviate it)

We agree with the reviewer and have now edited the text to include the full spelling of "quasi-mesenchymal" instead of the abbreviation "QM."

ductal-NE (while NE is commonly used, since 'ductal' is spelled out, is it necessary to abbreviate NE? It would be easier to read throughout by talking about a "neuroendocrine phenotype" too.)

We have edited the text to reflect the change from "NE" to "neuroendocrine" throughout the text.

Reviewer #2 (Remarks to the Author):

In this interesting and provocative manuscript by Dr. Sears and colleagues, the authors propose that deregulated c-Myc promotes a NE differentiation program stochastically during PDAC development, leading to worsened disease outcomes in patients and gemcitabine resistance. This is an addition to the field, and opens new avenues for consideration. Consideration to a few points will strengthen the manuscript for the field.

1. The NE phenotype – is it an association or is it causal in the features described here? For example, both EMT and Sox2 expression are discussed as occurring in the same PanIN/PDAC cells, but the authors are not claiming that EMT or Sox2 expression are driving the NE phenotype. Can the authors please clarify this topic in their manuscript.

We believe this is not causal; rather there is an association between the NE phenotype and stemness/EMT. We view the presence of these molecular features as indicative of an underlying cellular plasticity that supports aggressive behavior and therapeutic resistance. We have now commented on this in the discussion.

2. Gemcitabine resistance – is this a pharmacological or pharmacodynamic resistance mechanism in the cells/models used in this paper?

*We believe this is due, at least in part, to a pharmacodynamic resistance mechanism. In support of this, we examined our RNA-sequencing data from HPAFII cells treated with gemcitabine vs PBS for genes involved in gemcitabine resistance. We found that while the gemcitabine transporters (SLC29A1 and SLC29A3) and genes involved in gemcitabine activation (DCK and CMPK1) remained unchanged, gemcitabine treatment significantly increased cytidine deaminase (CDA), an enzyme that inactivates the majority of intracellular gemcitabine. Additionally, while we would expect genes known to be regulated by gemcitabine to be reduced upon treatment, we observed that RRM1, RRM2 were unchanged, while RRM2B was actually significantly increased by gemcitabine treatment (**new Supplementary Figure 3b**). We have commented on these findings in the text.*

3. Myc knockdown – this is known to be lethal in a cell autonomous fashion so the last experiments with this in human PDAC cells are not very informative. The alternative, of suppressing one copy of Myc in vivo using mice (as Owen Sansom has done in the colon) would be informative regarding the

percentage of NE cells in PanIN/PDAC, as this is not lethal.

We agree with the reviewer. This data was included (in collaboration with Owen Sansom) in the original Figure 3a-b, Supp. Fig 2a and shows that loss of one copy of Myc in KPC mice significantly decreases the percentage of ductal lesion-associated SYP+ cells.

4. Is CK-SYP expression an independent risk factor for poor outcomes in PDAC patients, by a multi-variate analysis? Possibly, the small sample size used here will be large enough to assess this point; and in that case I suggest the authors collaborate with those who have large tumor banks and outcomes data (Andrew Biankin, for example).

Thank you for the suggestion. To determine if age, clinical stage or grade significantly altered the relationship between co-staining and recurrence, we included each of these factors in a multiple linear regression model. None of the variables significantly altered the magnitude of the effect of co-staining; therefore we have concluded that these are not confounding our analysis of CK-SYP co-staining and disease-free survival and that CK-SYP expression is an independent risk factor for poor outcomes in PDAC patients. We have now stated this in the text. Further, using the published RNAseq data from Andrew Biankin, we also include data demonstrating that neuroendocrine upregulated genes are enriched in the poor outcome squamous subtype they define (Figure 2A).

REVIEWERS' COMMENTS:

Reviewer #1 (Remarks to the Author):

I am very pleased with the changes made to the manuscript and I am happy to support its publication in Nature Communication.

Reviewer #2 (Remarks to the Author):

The authors have responded appropriately to my concerns and the MS is now suitable for final approval.